# Mechanisms underlying TARP modulation of the GluA1/2-γ8 AMPA receptor

Beatriz Herguedas [1,2,6], Bianka K. Kohegyi[1,6], Jan-Niklas Dohrke[1,3], Jake F. Watson [1,4], Danyang Zhang[1], Hinze Ho [1,5], Saher A. Shaikh [1], Remigijus Lape[1], James M. Krieger [1] & Ingo H. Greger [1✉]

AMPA-type glutamate receptors (AMPARs) mediate rapid signal transmission at excitatory synapses in the brain. Glutamate binding to the receptor's ligand-binding domains (LBDs) leads to ion channel activation and desensitization. Gating kinetics shape synaptic transmission and are strongly modulated by transmembrane AMPAR regulatory proteins (TARPs) through currently incompletely resolved mechanisms. Here, electron cryo-microscopy structures of the GluA1/2 TARP-γ8 complex, in both open and desensitized states (at 3.5 Å), reveal state-selective engagement of the LBDs by the large TARP-γ8 loop ('β1'), elucidating how this TARP stabilizes specific gating states. We further show how TARPs alter channel rectification, by interacting with the pore helix of the selectivity filter. Lastly, we reveal that the Q/R-editing site couples the channel constriction at the filter entrance to the gate, and forms the major cation binding site in the conduction path. Our results provide a mechanistic framework of how TARPs modulate AMPAR gating and conductance.

[1] Neurobiology Division MRC Laboratory of Molecular Biology, Cambridge, UK. [2] Institute for Biocomputation and Physics of Complex Systems (BIFI) and Laboratorio de Microscopías Avanzadas (LMA), University of Zaragoza, 50018 Zaragoza, Spain. [3] Universitätsmedizin Göttingen, Georg-August-Universität, 37075 Göttingen, Germany. [4] Institute of Science and Technology (IST) Austria, Am Campus 1, 3400 Klosterneuburg, Austria. [5] Department of Physiology, Development and Neuroscience, University of Cambridge, Physiological Laboratory, Cambridge, UK. [6] These authors contributed equally: Beatriz Herguedas, Bianka K. Kohegyi. ✉email: ig@mrc-lmb.cam.ac.uk

AMPAR operation underlies synaptic plasticity and their uniquely rapid gating kinetics shape the time course of synaptic transmission[1]. The conformational cascade leading to receptor activation is governed by the combination of pore-forming subunits (GluA1-4) and diverse auxiliary subunits, that are expressed in specific patterns throughout the central nervous system[2,3], but the sequence of this cascade is incompletely understood.

Electrophysiological, structural and single-molecule imaging studies have provided a framework for the AMPAR gating cycle[4–8], which is dictated by the architecture of the receptor. Two functionally distinct subunit pairs (termed AC and BD) form a four-fold symmetrical channel that is gated by an extracellular region (ECR) of two-fold symmetry. The ECR is comprised of the ligand-binding domain (LBD) and the N-terminal domain (NTD), both of which fold into bi-lobate 'clamshell' structures that are arranged as dimers of dimers[9,10]. Glutamate binding to the LBD triggers clamshell closure, which transmits to the transmembrane domain (TMD) to open the channel's gate, or leads to desensitization through rupture of one or both LBD dimers[11]. Both routes are believed to proceed in parallel from a short-lived, closed-clamshell transition state (Fig. 1a)[7], and are modulated by auxiliary subunits[5].

TARPs are principal AMPAR auxiliary subunits, comprised of three groups; Type-1a (γ2, γ3) and Type-1b TARPs (γ4, γ8) are widely expressed, generally slowing gating kinetics, increasing agonist potency and reducing channel rectification by polyamines[2,3]. The sequence-diverse Type-2 TARPs (γ5 and γ7) are less well studied[12,13]. Belonging to the claudin family, TARPs have an elaborate extracellular domain consisting of a five-stranded beta-sheet and associated loops, which transiently engage the highly dynamic AMPAR LBDs and LBD-TMD linkers to modulate gating through currently incompletely understood mechanisms[14–16]. Gating modulation also depends

on arrangement of the core subunits[17,18], and on TARP stoichiometry: up to four TARPs can bind at two pairs of non-equivalent binding sites (termed A'C' and B'D'), that are formed by the adjacent AMPAR M1 and M4 transmembrane helices[14,19–22]. The smaller Type-1a TARPs (as well as cornichon subunits) can locate to both sites[14,17,23–25], while Type-1b TARPs (and GSG1l; germ-cell specific gene 1-like protein) preferentially associate with the spatially more accessible B'D' sites (Fig. 1; right panels)[24,25]. The rules underlying these distinct associations patterns are not resolved. These rules will also determine the reach of the TARP loops for the LBDs, and therefore centrally contribute to AMPAR modulation. With an array of auxiliary subunits interacting at multiple sites, a plethora of AMPAR combinations are possible[26]. In forebrain neurons GluA1/2 heteromers containing two TARP-γ8 subunits at their B/D sites constitute a major AMPAR combination[27,28], with variable possible interactions at the A'C' sites[26,29,30].

In this work, we use a combination of cryo-EM, all-atom MD simulations, and patch clamp electrophysiology to study GluA1/2 regulation by TARP-γ8. We observe gating-state-specific contacts of the extracellular γ8 loops with the LBDs and gating linkers, and reveal how a cytosolic interaction between the γ8 TM4 helix with the pore helix of the selectivity filter shapes channel rectification. Receptor activation leads to electrostatic potential changes around the gate that are enhanced by the γ8 β4-loop, which may funnel cations to the pore entrance. Furthermore, a concerted widening of the pre-M1 and M3 helices is amplified by the TARP, and is coupled to the selectivity filter through the Q/R site, impacting conductance. Lastly, we show that the Q/R site forms a major binding site for permeating ions. Together, we provide a glimpse of how auxiliary subunits enhance permeation, and open multiple avenues for further study of the complex interplay between AMPARs and their principal auxiliary subunits.

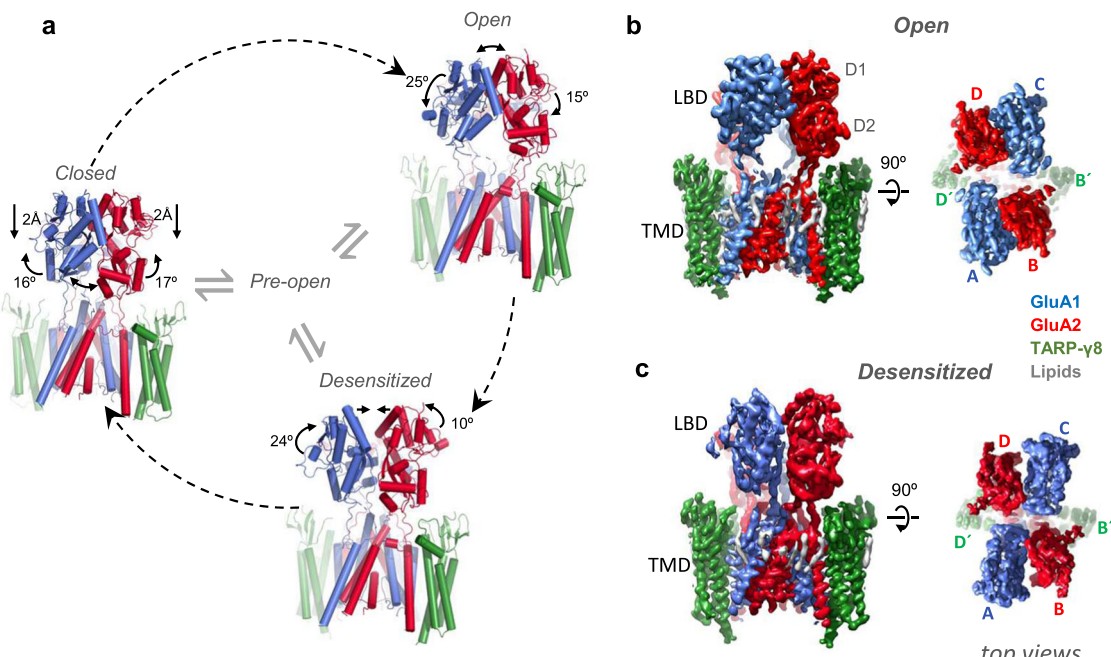

**Fig. 1 Cryo-EM structures of desensitized and activated GluA1/2_γ8 receptors. a** Periphery: simplified AMPAR gating cycle using models of closed (resting; PDB: 6QKC), open and desensitized states. Labels (values and arrows) indicate conformational changes leading to the next state along the stippled arrows (e.g. labels on the closed state reflect the changes leading to the open state, etc.). Centre: current AMPAR gating model[7] where a pre-open transition state (centre) leads to either open or desensitized conformations. **b** Cryo-EM map of the open state; side view left, top view right. Transmembrane domain (TMD) and ligand-binding domain (LBD) layers are indicated; D1 and D2 denote upper and lower LBD lobe, respectively. Subunit labels are denoted (A–D and B'D'). **c** As in (**b**) but for the desensitized state.

## Results

**Trapping GluA1/2_γ8 in open and desensitized states**. We expressed GluA1 together with a GluA2_γ8 fusion construct in Expi-HEK293 cells, resulting in a GluA1/2 heteromer associated with two TARP-γ8 subunits, the preferred stoichiometry for this TARP[17,25,27]. To trap the complex in the open conformation, we exposed the sample to a saturating concentration of L-glutamate (10 mM) together with cyclothiazide (CTZ; 100 μM) prior to vitrification; CTZ blocks desensitization by locking LBD dimers in an active conformation[11]. Cryo-EM data processing (Supplementary Fig. 1) indicated that not only open states (L-Glu+CTZ bound) but also desensitized receptors (L-Glu only) were present on the same EM grid, enabling a direct comparison of both states after classification and refinement procedures. 3D reconstruction resulted in the classic three-layered AMPARs, and further focused refinement and classification of the LBD and TMD layers separately produced maps with resolutions at ~3.5 Å for both open and desensitized states (Fig. 1b, c and Supplementary Figs. 1–3; Supplementary Table 1). Together with our resting-state structures (PDB: 6QKC; 7OCD)[17,25], this permitted an in-depth comparison of three conformational states.

**Gating transitions in the LBD layer**. Agonist-triggered closure of an LBD clamshell gives rise to an unstable, 'pre-active' state from which AMPARs open or desensitize (Fig. 1a, centre)[7]. Activation requires an intact interface between the upper (D1) lobes of an LBD dimer, and separation of the lower (D2) lobes upon clamshell closure transmits gate opening through the LBD-TMD gating linkers[14,22]. Desensitization, on the other hand, involves D1 interface rupture (Fig. 1a, c, and Supplementary Fig. 4a, b), which relieves LBD tension on the linkers to close the gate (Supplementary Movie 1)[11]. These LBD conformations were apparent during cryo-EM data processing, aiding classification of receptors with either separated D1 lobes (desensitized) or with separated D2 lobes (activated) (Fig. 1a–c and Supplementary Fig. 1c).

The 'upward' pull of the D2 lobes on receptor activation causes an increase of the D2 inter-lobe distance from 17.3 Å in the resting state to 31.4 Å (between GluA1 S631 and GluA2 S635); this distance is comparable between resting and desensitized states (17.3 vs 15.0 Å). In the desensitized state, D1 interface rupture separates the LBD dimer by 28 Å between GluA1 Ala737 and GluA2 Ser741, but is mostly unchanged between resting and open state (17.7 vs 17.4 Å). These values are comparable to homomeric GluA2 receptors associated with four TARP-γ2 or with two GSG1l subunits[14,24], and even to isolated LBDs[11]. Therefore these global motions are mostly independent of auxiliary subunit type and stoichiometry, which are expected to influence rate constants between these transitions.

As outlined below, LBD motions will be influenced by core subunit positioning—GluA1 preferentially locates to the AC positions and GluA2 to the BD positions (Fig. 1b, c), resulting in their differential impact on gating (Supplementary Fig. 4)[17]. To capture LBD gating motions, we determined angular displacements of all Cα atoms between two states relative to a centre of rotation, and computed mean values (see 'Methods'). In the resting-to-open transition, upward motion of the D2 lobes for GluA1 and GluA2 (16.0° ± 0.5° and 17.0° ± 0.5°, respectively), relative to the centre of rotation of an LBD dimer, is accompanied by a 2.2 Å vertical compression of the LBD layer towards the membrane. A slightly increased closure of the GluA2 cleft has been observed previously[14], and likely relates to the gating dominance of subunits in the BD position[17,18]. Evidently, the GluA2 D2 lobes experience fewer spatial constraints than the GluA1 D2 lobes, which interact not only with the GluA2 LBD of

the adjacent dimer but also with the diagonally opposed GluA1 subunit, potentially limiting their conformational freedom (Supplementary Fig. 4c, d).

The open-to-desensitized transition is accompanied by a large rotation of the GluA1 LBDs while GluA2 undergoes smaller, more local rearrangements (average residue movements of 24.5 ± 0.7° for GluA1 versus 14.7 ± 0.6° for GluA2). This is due to the GluA1 D1 lobe driving most of the D1 interface rupture upon desensitization in addition to closure of the D2 interface, leading to a rotation of the whole GluA1 LBDs (Supplementary Fig. 4a, b; Supplementary Movie 1). Surprisingly, the GluA2 D1 lobes move very little while the GluA2 D2 lobes still rotate towards the GluA1 D2 lobes. A similar asymmetry is also seen in the return of the desensitized dimers back to the resting state conformation with the GluA1 D1 lobes moving more to bring them back together (GluA1 23.4 ± 0.7° versus GluA2 9.5 ± 0.5°). These subunit-specific LBD rearrangements will also influence their interaction with the extracellular TARP loops and, in turn, gating of the AMPAR (Supplementary Fig. 5a).

**State-dependent γ8 loop interactions with the LBD**. Three TARP loops are central to modulation: the 'β1' and 'β4' loops emerge from extracellular segment 1 (Ex1) that connects the TM1 and TM2 helices and the shorter Ex2 loop/segment bridges between the receptor-binding TM3 and TM4 helices (Fig. 2a; Supplementary Fig. 5b, c)[14–16,19]. These loops vary in sequence and length between TARPs. Due to their flexibility, interactions of the γ2 and γ8 loops with the LBDs are largely unresolved[9,17,25]. To gain insights into their conformational spectrum, we performed 3D classifications focusing on the LBDs and γ8 loop sector ('Methods'). Aided by all-atom MD simulations, this approach revealed β1 loop contacts on the LBDs (Fig. 2b, c and Supplementary Figs. 6, 7, 8b, c).

*β1 loop*. Compared to other TARPs, the β1 loop (connecting β-strands 1 and 2; Fig. 2a) is elongated in γ8, enabling a more versatile engagement with both the GluA1 and GluA2 LBDs. We previously showed that this γ8 loop readily reaches the distal (upper) D1 lobe of the GluA2 LBD (around Lys410), and that introducing an N-glycan at this D1 'acceptor site' (at Asn411) selectively reduced γ8's impact on desensitization entry[17]. The β1 loop is 12 residues shorter in γ2, and this GluA2 glyco mutation was of no functional effect in GluA1/2_γ2 complexes, likely due to its lower probability to reach this site[17].

At its base, the β1 loop is constrained through a cysteine bridge (Cys52–Cys91; Supplementary Fig. 5b, inset) unique to Type-1 TARPs, which will impact its dynamic range. Although loop density is not well defined in our final maps, focused 3D classification of the LBD-γ8-loop region provided sufficient signal to follow its path during gating (Supplementary Figs. 6, 7). In the desensitized state, a predominant 'elongated' β1 conformation facilitates robust interaction with the GluA2 D1 lobe, proximal to Lys410 (Fig. 2b; Supplementary Fig. 6b). Interestingly, this contact was mostly absent in the active receptor and was diminished in a resting state structure ('Methods'; Supplementary Fig. 6c, d). In resting and active states, the β1 loop adopted a 'collapsed' conformation, favouring engagement of the GluA1 and GluA2 D2 (lower) LBD lobes (Fig. 2b). β1 loop contacts with GluA1 D2 predominated in the active receptor, while interactions with both GluA1 and GluA2 D2 lobes, were evident in the resting state.

To further investigate this state-dependent β1 behaviour, we performed focused classification of the *entire* Glu/CTZ data set (as detailed in Supplementary Fig. 6, 7), which in addition to open and desensitized particles also contained particles excluded in our

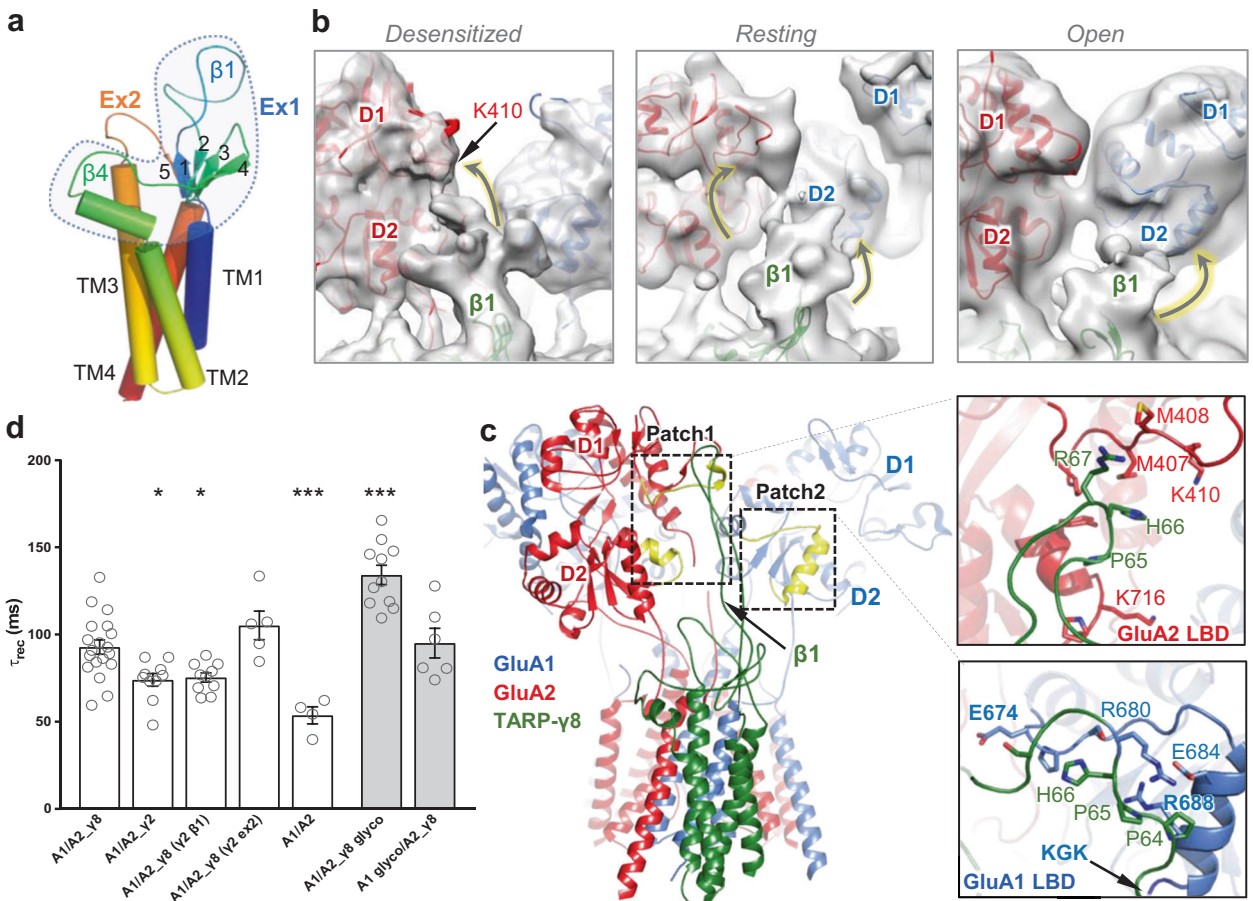

**Fig. 2 TARP loop interaction sites on the LBDs. a** Schematic of TARP γ8, showing transmembrane helices (TM1–4), the 5-stranded beta sheet and the three attached loop elements. **b** cryo-EM maps obtained by focussed processing, showing state-dependent interaction of the β1-loop with the GluA1 and GluA2 LBDs (indicated by arrows). **c** Left panel: β1-loop contacts on the LBDs obtained by MD simulations reveal two patches (yellow) that closely resemble the interactions sites described in (**b**). Right panel: environment of patch 1 on the GluA2 LBD (red), and the GluA1 LBD (blue). **d** The TARP-γ8 β1 loop mediates slowing of recovery from desensitization. Functional interaction between GluA2 D1 lobe and γ8 β1-loop may contribute to the stabilization of the desensitized state. Desensitization recovery tau (ms): A1/A2_γ8, 92.79 ± 4.09 (mean ± SEM), $n = 19$; A1/A2_γ2, 74.01 ± 3.66, $n = 10$, (*$p = 0.0210$); A1/A2_γ8 (γ2 β1), 75.38 ± 2.62, $n = 10$, (*$p = 0.0381$); A1/A2_γ8 (γ2 ex2), 105.2 ± 8.20, $n = 5$ (ns, $p = 0.5056$); A1/A2, 53.62 ± 4.90, $n = 4$, (***$p = 0.0002$); A1/A2_γ8 glyco, 134.2 ± 5.51, $n = 11$, (***$p < 0.0001$); A1 glyco/A2_γ8, 95.10 ± 8.48, $n = 6$, (ns, $p = 0.9996$). One-way analysis of variance (ANOVA) with Dunnett's multiple comparison test (against A1/A2_γ8 as control), $p < 0.0001$. Source data are provided as a Source data file.

initial processing scheme (Supplementary Fig. 1c). Classes with the GluA2 D1/β1 interactions consistently exhibited desensitized LBDs, while classes with GluA1 D2/β1 interactions (or showing no clear contacts) were consistently open state dimers (Supplementary Fig. 6a). Indeed, 3D refinement of the 'GluA2 D1/β1 particles' generated a desensitized receptor conformation (Supplementary Fig. 7c). Although still state-dependent, interactions between GluA1 D2 and the β1 loop are less stable and accordingly signal density was poorer in the open state model (Supplementary Fig. 7d).

To probe the functional impact of this loop, we swapped β1 from TARP-γ2 into γ8, reasoning that the shorter γ2 loop is less likely to reach the GluA2 D1 lobe. The loop chimera had no impact on desensitization entry or the equilibrium current but selectively affected recovery from desensitization (Fig. 2d, and Supplementary Fig. 8a). Unlike other TARPs, γ8 slows recovery of GluA2-containing AMPARs from desensitization[17,31] (but not of GluA1 receptors[32,33]). This behaviour was blunted by exchanging the β1 loop for that of γ2, with the loop chimera exhibiting γ2-like recovery kinetics. Moreover, this effect was specific to β1 as swapping the TARP Ex2 loops between γ2 and γ8 fully retained γ8's slow recovery phenotype. We hypothesize that

the γ8 β1-D1 contact stabilizes the desensitized state, by slowing recovery out of this conformation. Furthermore, while speeding desensitization entry to levels seen with TARP-free receptor[17], the aforementioned GluA2 D1 glyco-mutant (harbouring a glycan at Asn411[17]) slowed recovery even further (Fig. 2d), suggesting that the glycan stabilizes the desensitized LBD conformation. Introducing the glycan into GluA1 D1 was of no functional consequence (Fig. 2d), further supporting a specific relationship between β1 and the GluA2 subunit.

To explore these findings further we performed MD simulations of a resting-state structure (PDB: 6QKC) and our open state structure, lasting 350 ns each. The resting state simulations revealed two main β1 interaction hotspots on the LBD (Patch 1 and 2) (Fig. 2c, yellow; and Supplementary Fig. 8b, c). Patch 1 includes GluA2 D1 residues close to Lys410 and a helix in D2 immediately beneath, around Lys716. In this configuration β1 effectively wedges between the D1 and D2 lobes (Fig. 2c, upper inset), closely resembling the resting state cryo-EM structure (Fig. 2b and Supplementary Fig. 6d). Patch 2 comprises residues around Glu674-Leu700 in the GluA1 D2 lobe (Fig. 2c). Here, β1 also comes into close proximity to the GluA1 KGK motif (Lys693-Lys695) a region strongly implicated in TARP modulation, but

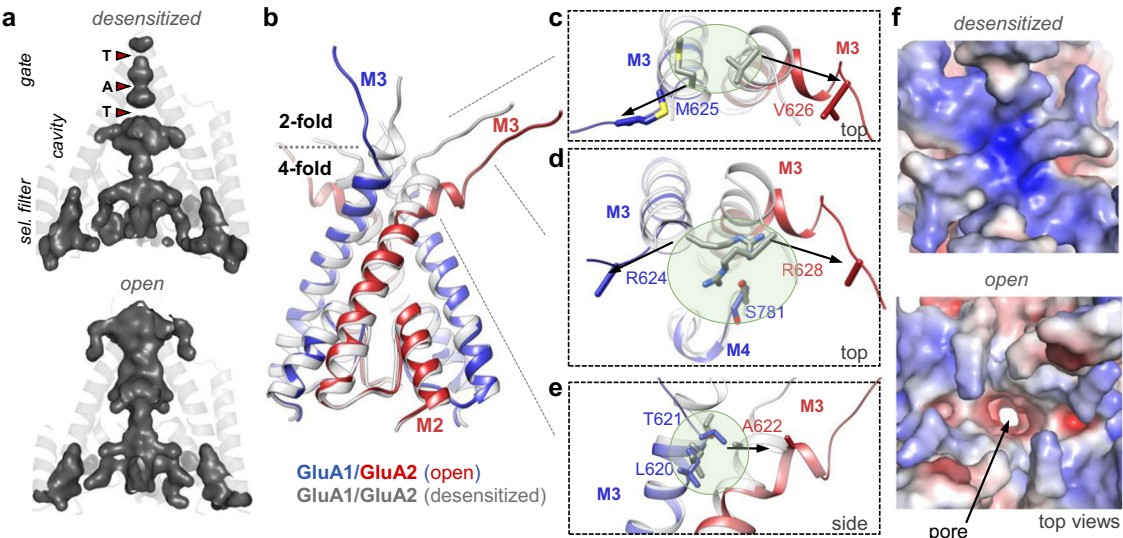

**Fig. 3 Gate transitions to the open state. a** Volume of ion conduction path for desensitized (top) and open states (bottom). The three constriction points on the M3 gate (T–A–T) are indicated. Of note are the gate-flanking negative modulator (GYKI, Perampanel) binding sites in the open state volume. **b** Models of the gating core (M2 and M3 helices) of the desensitized (grey) and open states (colour). The two M3 helices in the front belong to a heterodimer. Unzipping of the 2-fold symmetric linker vestibule leads to M3 unwinding and opening of the M3 constrictions (4-fold symmetric). **c, d** Contacts rupturing in the 2-fold symmetric vestibule. **e** Contacts rupturing in the 4-fold symmetric gate. **f** Electrostatic surface potential of the desensitized (closed) gate entrance top and active (open) gate bottom (APBS range: $-10$ kTe$^{-1}$ to $+10$ kTe$^{-1}$).

through the TARP Ex2 segment[14,34]. In simulations of the open state structure, β1 mainly contacts Patch 2 on the GluA1 D2 lobe, including the KGK motif (Supplementary Figs. 6c, 7c); interactions with GluA2 were nearly absent, mirroring the cryo-EM data. We note that, with the resting state model, we also observed signs of more extended β1 conformations, characteristic of the desensitized state, but these were not stable under the nanosecond timescales of our simulations. Therefore, despite its versatile range, β1 has preferred LBD contact points, which are likely to be influenced by gating-state dependent LBD orientations (Supplementary Fig. 5a).

*β4 and Ex2-loops.* With γ8 docking to its preferred B'D' position in the GluA1/2 receptor[17,25], its β4 loop exclusively engages GluA2. We find that this interaction involves acidic residues in β4 (around Asp109 and Asp111) and the KGK motif Lys697–Lys699 in the GluA2 D2 lobe. TARP γ8 thereby engages this functionally critical motif[15,34,35] on both subunits: GluA1 via the β1-loop, and GluA2 through the β4-loop; the β1 interactions are likely to be unique to γ8. These versatile, state-dependent interactions may explain the complex effects of mutating the AMPAR KGK motif[15,34,35].

Upon receptor activation the M1 and M3 linkers bundle up and splay towards both the β4 and Ex2 loops of γ8, bringing these loops into contact with Gln508 in the M1 linker (Supplementary Fig. 5c). This transition may enable TARP control of the critical M3 gating linker and increase stability of the open state through tension on the M3 gating helices. Contacts between Ex2 (Tyr199, Tyr201) and the GluA1 M1 helix (Y519, E520), a region targeted by γ8-selective modulatory drugs[36], are maintained in both open and desensitized states.

**Gating transitions in the TMD sector.** The conduction path of the desensitized receptor closely mirrors the resting state with the gate shut (Supplementary Fig. 9a). We therefore used the desensitized structure to closely analyse the transition to the open state. The M3 gate adopts four-fold symmetry and blocks cation flux at three known constriction points, Thr–Ala–Thr (GluA1: 613–617–621; GluA2: 617–621–625) (Fig. 3a and Supplementary

Fig. 10a), whose side chains project toward the pore axis[14,22]. On the other hand, the vestibule above the gate adopts two-fold symmetry, and rearrangements in this region precede opening of the gate (Supplementary Movie 2); the force transmitted from LBD cleft closure unzips three sets of contacts between the M3 helices of a GluA1/2 dimer (Fig. 3b–e): (1) between GluA1 Met625 and GluA2 Val626 (Fig. 3c); (2) between GluA1 Arg624, GluA2 Arg628 and GluA1 Ser781 (Fig. 3d); and (3) contacts between GluA1 Leu620 and Thr621 and GluA2 Ala622 (Fig. 3e). Loss of these interactions leads to unwinding of the M3 C-termini, and rupture of the gate at GluA1 M625 and the Thr–Ala–Thr motif. These rearrangements in the vestibule are expected to determine activation kinetics, akin to NMDA receptors[37]. They will also alter electrostatics at the pore entrance: In desensitized and resting states, Arg624 (GluA1) and Arg628 (GluA2), contribute to a mostly positive potential, while rupture of these interactions and outward motion of the M3 linkers gives rise to a more negative surface potential (Fig. 3f). This potential change may attract cations to the pore entrance. Moreover, the TARP-γ8 β4 (acidic) loops further contribute to this: together with the M1 and M3 linkers, they generate a negatively charged path leading to the pore (Supplementary Fig. 9c, d). Hence, in addition to their modulation of channel kinetics, TARPs may also facilitate cation attraction in the activated receptor, and thereby contribute to the increased channel conductance[38].

Driven by their LBD trajectories (Supplementary Fig. 4b–d), the GluA2 M3 helices splay open more widely on activation than GluA1 and kink at Ala621 to engage the pre-M1 helices (Fig. 3b). The pre-M1 helices form a fence-like structure around the gate[10], and have to accommodate gate dilation (Supplementary Fig. 11). This asymmetry will affect AMPAR pharmacology, as kinking of the GluA2 M3 helices blocks a binding site for negative allosteric modulators (NAMs), including GYKI and perampanel[39–43]. This binding pocket remains largely intact on the GluA1 'sides', implying that access of these modulatory drugs is state-dependent and side-specific (Supplementary Fig. 9b).

Dilation of the M3 helices continues midway into the conduction path to GluA1 Ile609 and to the GluA2 equivalent,

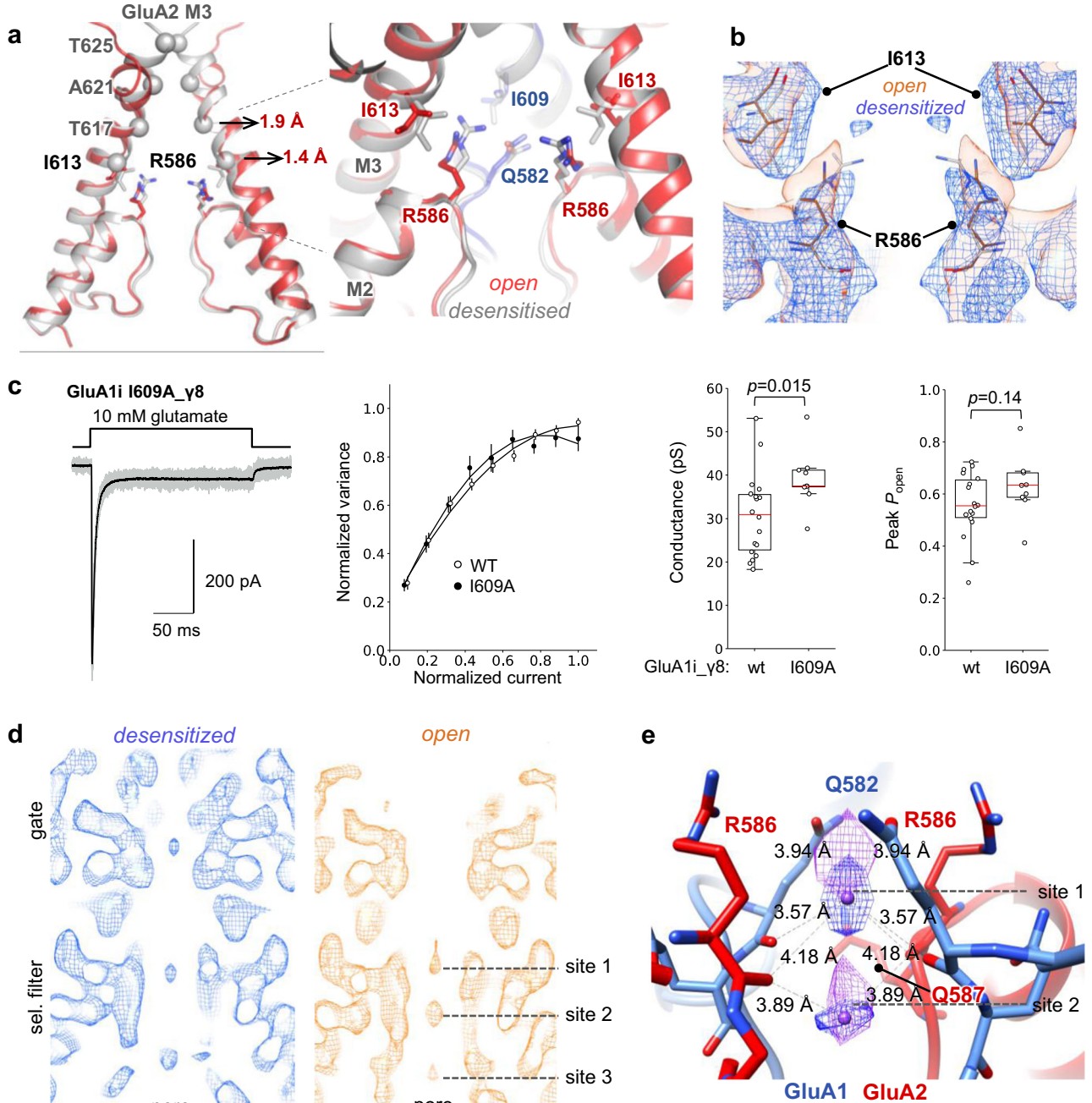

**Fig. 4 Relationship between M3 gate and Q/R ion binding site. a** Overlay of the gating cores (grey: desensitized; red: active), showing the GluA2 M3 constriction points (spheres) relative to the selectivity filter. Values of M3 dilation at Thr621 and Ile613 are depicted (in Å). Inset: Ile613 side chains are in van der Waals distance to the Q/R side chain Cγ,δ, the Q/R side chains form a polar constriction above the selectivity filter. **b** cryo-EM maps showing the relationship of Ile613 and Arg586 (desensitized map: blue; open map: orange). **c** Example current traces recorded from outside-out patches evoked by 200 ms pulse of 10 mM glutamate (20 individual responses in gray, average current in black). The plot in the middle shows normalised and averaged current-variance relationship for wild-type (open circles) and I609A mutant (closed circles) GluA1i_TARPγ8. The dashed lines are parabolic fits to the experimental points. Two boxplots on the right show single-channel conductance current peak, and open probability estimates for the wild-type and mutant receptor; the red horizontal line indicates the median; the boxes show the 25th and 75th percentiles; the whiskers enclose the points that fall within 1.5 times of the interquartile range. Source data are provided as a Source Data file. **d** Density maps revealing ion density in the selectivity filter (sites 1–3). Note the additional density peak in the gate of the desensitized map (blue). **e** Modelled $Na^+$ ions in density peaks, coordinated the Q/R site side chains and main chain carbonyls. Distances of ion coordination are shown and imply (at least partial) $Na^+$ hydration.

Ile613 (Fig. 4a). These isoleucines project towards the inner cavity, poised to restrict dynamics of the Gln582 (GluA1) and Arg586 (GluA2) side chains at the selectivity filter entrance (Fig. 4a, b and Supplementary Fig. 10b). GluA2 Arg586 is defined by RNA editing and is critical to AMPAR function, blocking

$Ca^{2+}$ flux, reducing channel conductance and limiting channel block by polyamines[44–47]. Interestingly, M3 helix widening on channel activation facilitates dilation of these critical residues away from the pore axis, in concert with Ile609 and Ile613 (Supplementary Movie 2), suggesting an interaction between the

gate and selectivity filter at this point (Fig. 4b). Shortening the M3 isoleucine side chain is expected to alleviate interaction with the M2 pore loop apex, primarily through the Q/R site residues, and to a lesser extent with GluA1/GluA2 residues F580/F584. Indeed, I609A, in a GluA1_γ8 tandem configuration, resulted in an increase of channel conductance, when assessed by non-stationary fluctuation analysis (NSFA; GluA1_γ8 wt $31 \pm 2$ pS, $n = 18$; GluA1_γ8 I609A $39 \pm 2$ pS, $n = 9$; Fig. 4c). This observation lends support to the idea that Ile609 spatially restricts dilation of the Gln582 side chains at the selectivity filter entrance, limiting the level of ion flux. No effect on open probability was apparent in the I609A mutant (Fig. 4c). Although attempted, the same analysis could not be reliably conducted for GluA2_γ8 due to its much lower single-channel conductance and atypical behaviour in NSFA[46,48]. These data reveal that opening of the M3 gate in AMPARs is coupled to the selectivity filter and influences channel conductance.

**Ion binding at the Q/R site.** Interaction of the Arg586 guanidinium groups with the GluA1 Gln582 side chains and the Met581 main chain generates a polar constriction for cation passage above the selectivity filter (Fig. 4a, inset). Beneath this constriction, the filter apex is lined with the Gln582 and Arg586 main chain carbonyls, separated by an inter-atom diameter of 6.5 Å, permitting the permeation of hydrated cations, as expected for a non-selective cation channel (Supplementary Fig. 10c)[49].

In the selectivity filter of the open channel two density peaks (sites 1 and 2) are evident in this region, likely corresponding to hydrated $Na^+$ ions, the most abundant cation in the preparation (Fig. 4d). A third density at Cys585/589 (GluA1/2) was also apparent in our earlier study under resting conditions[50], and may be due to multiple occupancies of the Cys residues. The upper peak (site 1) is coordinated by the GluA1 Gln582 side chains and the main chain carbonyls of Gln582 and Arg586, which project toward the pore axis and contribute to coordination of peak 2 (Fig. 4e). The distances between density peaks and coordinating protein atoms are again consistent with hydrated $Na^+$ ions[51]; ion binding at the Q/R-site adds to its established role in regulating cation permeation[44,45,52]. Similar densities are seen in maps processed with C1 symmetry and are further apparent in our recent GluA1/2 structures in complex with TARP γ8 and CNIH2[25] (Supplementary Fig. 10d). Smearing of the signal between sites 1 and 2 suggests partial occupancy, possibly caused by gating state-dependent widening of the Q/R constriction (Fig. 4b).

Densities also locate to the constrictions of the gate but these are only apparent in desensitized and closed state structures (Fig. 4d and Supplementary Fig. 10d), implying that these binding sites disappear upon gate opening. Together with state-dependent changes in conduction path electrostatics (Supplementary Fig. 10e), these data shine light on the control of ion passage through an AMPAR channel; when combined with the findings of the following section, they also suggest how auxiliary subunits enhance conductance.

**Impact of auxiliary subunit stoichiometry on activation.** The 2-fold symmetry in the gate region dictates that auxiliary subunits exert a different impact on gating, depending on their location to either the A′C′ site (formed by the $M1_{GluA2}$ and $M4_{GluA1}$ helices) or the B′D site (formed by $M1_{GluA1}$ and $M4_{GluA2}$) (Fig. 1b, c)[14,17,22–25]. We observe that activation leads to an asymmetric expansion of the pre-M1 region, occurring to a greater extent between the diagonally opposed GluA2 subunits (Supplementary Fig. 11a). This behaviour was also apparent in our recent GluA1/2 structures associated with both TARP-γ8 and CNIH2 subunits, and was accompanied by tilting and by counter-rotations of the

two auxiliary subunit pairs along their vertical axes[25]. Through their proximity to pre-M1, auxiliary subunit dynamics may thereby stabilize active state conformations. As the A′C′ positions are vacant in the GluA1/2 γ8 receptor (Supplementary Fig. 11b, c), a comparison with the CNIH2-lacking structures provided an opportunity to assess the impact of auxiliary subunit stoichiometry on AMPAR activation.

In the CNIH2-containing complex, the γ8-transmembrane helices undergo an anti-clockwise rotation (when viewed from the top), combined with a vertical tilt of the TARP[25]. A similar motion is seen in the CNIH2-lacking structures upon transition to the open state, suggesting that γ8 dynamics during AMPAR activation are independent of CNIH2. Interestingly, the gate-surrounding fence, including pre-M1 and the base of the M4 linker (M4L, at Leu783/787 in GluA1/GluA2), experienced a greater expansion when CNIH2 is present, compared to the CNIH2-free open structure (Supplementary Fig. 11b, c). This is evident when measuring distances between the diagonally opposed GluA2 Pro620 Cα atoms in pre-M1, and between the Leu787 Cαs in M4 (Supplementary Fig. 11a), where additional expansions of 0.8 and 0.9 Å are seen with CNIH2, respectively. The equivalent marker atoms in the GluA1 subunits expand to a lesser extent (0.5 and 0.3 Å more than in the CNIH-free structures).

The greater dilation of the gate-surrounding region in the presence of CNIH2 will facilitate separation between the M3 gate helices, which we observe: GluA2 Ala621 in M3, at the narrow constriction of the gate, expands by a further 3.3 Å in the CNIH2 complex, compared to the CNIH2-free open state (Supplementary Fig. 11c, d). A relationship between preM1 and M3 dynamics is also seen in our MD simulations of the open state, where expansion of the preM1 fence correlates with M3 gate widening (Supplementary Fig. 11e). We hypothesize that the pre-M1 segment is central to auxiliary subunit function.

Interestingly, it is not auxiliary subunit identity, but stoichiometry which may determine this regulation. Separation of preM1 in the activated GluA2 receptor associated with four TARP-γ2 molecules (PDB: 6DLZ) is comparable to the CNIH2-containing open complex, exhibiting similar dilations between opposing Ala621 and Ile613 residues. Therefore, occupation of all four binding sites facilitates maximal expansion of the gate-surrounding region. This in turn permits a wider separation of the M3 gating helix, and transmission down to the Q/R ion binding site to increase channel conductance (Fig. 4a–c)[2,3,38].

**TARP-γ8 contacts the pore helix to shape rectification.** TARP interactions with the AMPAR TMD impacts channel modulation, as demonstrated by γ8-selective drugs targeting the AMPAR-TARP interface[36,53,54], but contrary to the influence of the TARP loops[15,16,31], regulation by the TARP TMD is currently poorly understood. The interface between the TARP TM3 and TM4 helices and the GluA1 M1 and GluA2 M4 helices is well resolved (Fig. 5a and Supplementary Fig. 3, 12a). Analysis of our MD simulations[55] revealed interaction patterns between the two TARP helices with the receptor. While stable association between γ8 TM3 and the GluA2 M4 helix exist throughout their length, only the lower half of TM4 exhibited persistent interaction with GluA1 M1 (Fig. 5b). Interestingly, residues in the upper part of this helix, N-terminal of γ8 Phe212 (contacting GluA1 Phe527), form much weaker contacts. This observation divides the γ8 M4 helix into a loose upper part, which facilitates access of modulatory ligands[53,56,57], and a coupled lower part. One particularly stable interaction occurred between γ8 Val220 and Ile569 (Fig. 5b), which resides at the base of the GluA1 M2 pore helix (Fig. 5c), providing a direct link between γ8 and the AMPAR gating machinery; this interaction also involves GluA1 Phe537 in M1.

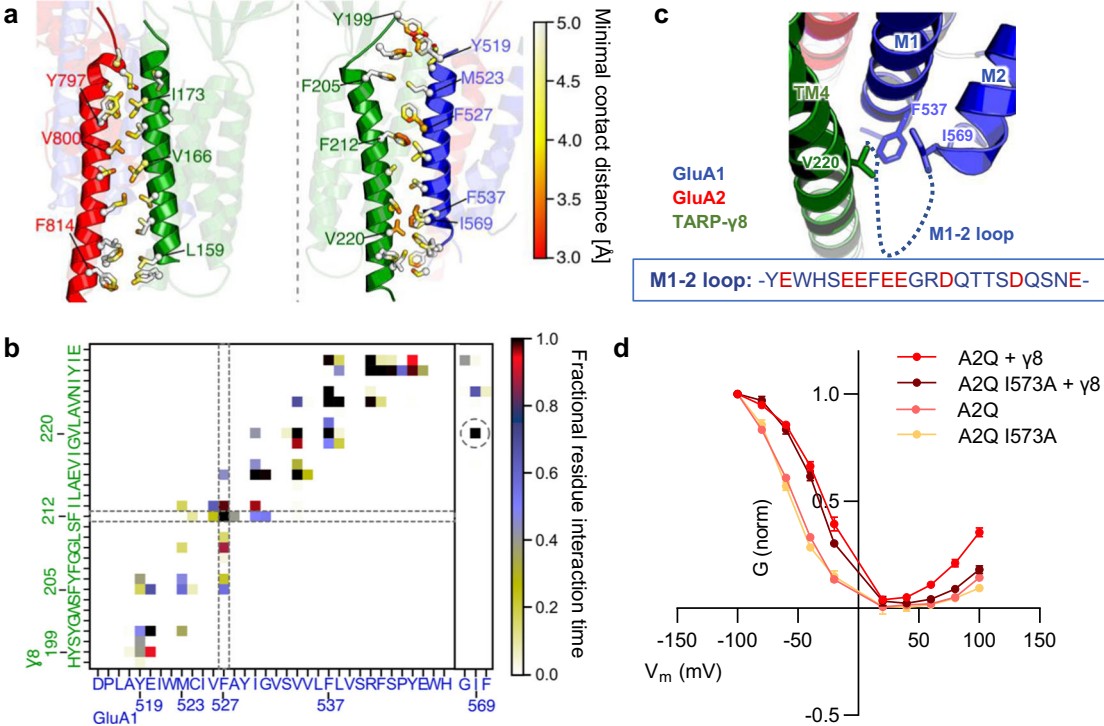

**Fig. 5 TARP coupling to the M2 pore loop controls rectification. a** Quantification of contact points between the TARP-γ8 TM3 and GluA2 M4 helix (left) and TARP-γ8 TM4 with the GluA1 M1 helix (right). **b** Contact point analysis using CONAN, revealed a stable interaction between Val220 (in TARP-γ8 TM4) and Ile569 (GluA1 M1). **c** Zoom into the Val220/Ile569 interaction regions, also involving F537 on GluA1 M1. The stippled line indicates the GluA1 M1-M2 cytoplasmic loop, which is currently unresolved and is highly charged (sequence insert). **d** Conductance/voltage (G/V) plot revealing a selective, TARP-γ8-dependent impact of GluA2 Ile573Ala on outward rectification (mean ± SEM, $n = 10$–13). Source data are provided as a Source data file.

When mutating the residue equivalent to Ile569 in homomeric GluA2 (Ile573) to alanine, gating kinetics of the mutant were unaffected both in the presence or absence of TARP-γ8 (Supplementary Fig. 12b). However, channel rectification, which is caused by polyamines binding to the negatively charged cytoplasmic pore entrance[51,58,59], was altered (Fig. 5d, Supplementary Fig. 12c). TARPs alleviate polyamine block and increase channel conductance through incompletely resolved mechanisms[59]. Mutation of Ile573 to alanine, which will uncouple the M2 pore helix from the TARP, altered GluA2 rectification specifically at positive membrane potentials, preventing γ8 from modulating current flow. As evident from conductance-voltage (G–V) plots (Fig. 5d), channel conductance of the mutant at negative potentials was unaffected, both in the presence and absence of γ8. However, at positive potentials, the conductance profile in the presence of γ8 was akin to receptors lacking auxiliary subunits. This mutation therefore has a selective impact on outward current flow.

Polyamines, which bind at the cytoplasmic pore entrance (at GluA1 D586/GluA2 D590[60]), will be most tightly associated at positive potentials, as the membrane potential will drive their transit through the channel[58,61]. We hypothesize that TARP interaction with the base of the M2 pore helix stabilizes M2, and impacts conformations of the highly negatively charged cytoplasmic loops connecting the GluA1 M1 and M2 helices (Fig. 5c). These loops are ideally positioned to engage positively charged polyamines and may be constrained away from the pore through interactions with the TARP C-terminus. Indeed, receptor activation induces the tilting of auxiliary subunits towards the cytoplasmic pore entrance further enabling this interaction[25]. Lipid molecules concentrate at this critical region (Supplementary Fig. 12d), and through their bridging between γ8 and the AMPAR M2/M3 gating core, they are ideally positioned to contribute to this regulation.

## Discussion

Enabled by their architecture, TARPs regulate multiple components of AMPAR signaling through versatile interactions, ultimately increasing the magnitude and duration of charge transfer, tuning synapse output[2,3,29,62–64]. Activation not only leads to rearrangements of the LBD layer[4,6,65], but also of TARPs: the TARP-γ8 cytoplasmic portion tilts towards the channel pore axis while the extracellular β-sheet bends away from this axis[25], impacting the means of their engagement with receptor LBDs. LBD interactions will be determined by arrangement of the core subunit pairs[17], and will be particularly versatile for the β1-loop, which is uniquely elongated in γ8 (Fig. 2b, c). Dictated by γ8's preferential association with the B′D′ sites (Fig. 2)[25] this loop impacts the desensitized state through engagement of the GluA2 LBD upper lobe. This contact appears to slow desensitization recovery, a process requiring opening of the LBD clamshell to release L-glutamate from its binding site[7]. Of note, recovery kinetics differ greatly in AMPAR homomers[31,33], and so does the effect of γ8 on these rates, speeding the recovery of GluA1 but slowing that of GluA2 homomers, and of GluA2-containing heteromers[17,31–33]. These differences will lead to unique regulation of GluA1 homomers, a $Ca^{2+}$-permeable AMPAR implicated in LTP[66], and in various diseases[67]. It remains to be established how β1-interactions with the lower lobes of the GluA1 (prominent in activate states) and GluA2 (prominent in resting states) LBDs modulate the receptor, and how loop dynamics stabilize the closed-cleft state[5,68]; our data provide a roadmap to assess this in future studies.

A disordered β1-loop appears to be a characteristic of Type-1 TARPs. Type-2 TARPs (γ5 and γ7) by contrast are predicted by AlphaFold[69] to form a hairpin with extended beta strands 1 and 2. This structural difference is most likely due to the absence of the Cys52–Cys91 cysteine bridge in Type-2 TARPs (Supplementary Fig. 5b), and is expected to reduce conformational freedom of this

segment, impacting its dynamic range. Other essential modulatory segments, the β4 and Ex2 loops, are also predicted to adopt different conformations in γ5 and γ7: lack of a kink in their TM2 helices leads to a reorientation of the β4 acidic loop, while the Ex2 loop is rigidified due to both an extended beta strand 5 and the TM3 C-terminus. These features will contribute to the different modulatory properties of Type-2 TARPs[12,13].

At the base of the receptor, van der Waals contacts between γ8 TM4 (at Val220) and the M2 pore loop (Ile569) alleviate outward rectification (Fig. 5c, d), shedding light on how TARPs modulate AMPAR-block by intracellular polyamines[59]. This interaction, which involves annular lipids (Supplementary Fig. 12d and ref. [25]), could stabilize the negatively charged AMPAR M1-2 cytoplasmic loops away from the pore axis, thereby reducing polyamine binding at the pore entrance. 'Trapping' of the M1-2 loops away from the pore is likely aided by tilting of TARP (and CNIH2) cytosolic elements toward the pore axis on activation[25]. In line with these observations, a recent study reported a role for the M1-2 loops in rectification of kainate receptors[70].

Our data provide insights into TARP modulation of conductance. First, we propose that the increase in negative electrostatic potential, contributed by gating linker rearrangements and the TARP β-4 acidic loops, facilitate cation attraction to the pore (Fig. 3f, and Supplementary Fig. 9c, d). Within the conduction path densities consistent with permeating $Na^+$ ions locate to the constrictions of the M3 gate in closed states and accumulate at the Q/R site in both resting and active receptors (Fig. 4d; and PDB: 7OCE and 7OCF). Cation-binding at the Q/R site is in line with a recent MD simulation study, which identified this region as a major binding site for $K^+$ and $Na^+$ ions in the pore[71], and with a recent kainate receptor structure[70]. The diameter between the Q/R site main chain carbonyls (6–7 Å) is consistent with non-selective permeation of hydrated (mono- and di-valent) cations. This site is coupled to the gate through isoleucine residues (Ile609 and 613) in the M3 gating helices, that form a hydrophobic ring above the Q/R site. It is conceivable that widening between the Q/R side chains on M3 gate-dilation alters binding of permeating cations and thereby impacts channel conductance (Fig. 4 and Supplementary Fig. 10b, c).

The gate-surrounding pre-M1 and M4 linker region has been recognized as a key gating element across iGluRs[72,73]. We propose that this segment constitutes a major control point for auxiliary subunits. Activation-triggered auxiliary subunit rearrangements enable widening of the pre-M1 fence (Supplementary Fig. 11b, c), facilitating a wider separation of the M3 helices, and ultimately the selectivity filter entrance to promote ion flux. MD simulations predict that motions of the pre-M1 and M3 helices correlate (Supplementary Fig. 11e). How M1 and M3 linker tension on LBD cleft closure orchestrate expansion of the pre-M1 and M3 helices is an intriguing open question. Moreover, the nature and dynamics of annular lipids concentrating in this region[17,23,25] might further influence preM1 dilation, and couple this element to the M2 and M3 helices of the conduction path (Supplementary Fig. 12d)[25]. At the angstrom-scale of these molecular machines, only small movements are required to substantially alter the flow of ions across the membrane, to ultimately impact synaptic computations. With such an array of auxiliary subunits having developed for the AMPAR[2,3,26,29,74], the specific nature of this regulation must be of crucial importance for brain function. Our structures begin to reveal the precise means by which these proteins exert their essential influence to ultimately shape synaptic transmission and plasticity.

## Methods

**Constructs**. All constructs were produced using IVA cloning[75] and have been previously described[17]. GluA1 cDNA (rat, flip) was cloned into the pRK5 plasmid, and a FLAG tag was added at the N-terminus (A1_FLAG plasmid). GluA2 cDNA (rat

cDNA sequence, flip, R/G, Q/R edited) was fused with a GGSGSG linker to TARP γ8 (synthetic gene, rat protein sequence), a human rhinovirus 3C (HRV 3C) protease site and an eGFP; and cloned into pRK5 (A2_γ8_eGFP plasmid)[17].

For electrophysiology experiments, TARP-tandem and mutation constructs were produced on pRK5 plasmids expressing rat wildtype GluA1 (flip) and GluA2 (flip, R/G, Q/R edited) cDNAs. Tandem constructs were generated by connecting the C-terminus of GluA1 (excluding residues Ile836-Leu889) or GluA2 (excluding residues Gln840-Ile862) with a GGSGSG linker sequence to rat TARP γ2 (Glu2-Arg319) or γ8 (residues Glu2-Lys419; 4 amino-acid deletion). For non-tandem recordings, GluA2 (flip, R/G edited, Q/R unedited) and TARP γ8 were expressed from pIRES vectors, with coexpressed EGFP and mCherry, respectively (1:2 ratio for transfection). The TARP γ8 β1-loop chimera was constructed by replacing γ8 residues Leu50-Leu82, with the equivalent region of γ2 Val39-Met58.

**Electrophysiology**. Human embryonic kidney (HEK) 293T cells (ATCC: Cat# CRL-11268, RRID: CVCL_1926) were cultured at 37 °C and 5% $CO_2$ in DMEM (Gibco; high glucose, GlutaMAX, pyruvate, Cat# 10569010) supplemented with 10% foetal bovine serum (Gibco) and penicillin/streptomycin, and transient transfection was achieved using Effectene (Qiagen). Transfected cells were plated on poly-L-lysine-coated glass coverslips and were incubated with media containing 30 μM 2,3-dioxo-6-nitro-1,2,3,4-tetrahydrobenzo[f]quinoxaline-7-sulfonamide (NBQX; Tocris) to mitigate AMPAR-mediated toxicity.

Outside-out patch clamp recordings were performed by applying L-glutamate (1 mM) using a double-barrel fast agonist application system mounted on a piezoelectric translator (Physik Instrumente). Data were acquired using a MultiClamp 700B amplifier (Molecular Devices) and digitized using a Digidata 1440A digitizer (Molecular Devices). Recordings were analysed with Clampfit (Molecular Devices). The extracellular solution consisted of (in mM): NaCl (145), KCl (3), $CaCl_2$ (2), $MgCl_2$ (1), glucose (10), and HEPES (10), adjusted to pH 7.4 using NaOH. The intracellular solution contained (in mM): CsF (120), CsCl (10), EGTA (10), ATP-sodium salt (2), HEPES (10), and spermine (0.1), adjusted to pH 7.3 with CsOH. Borosilicate glass electrodes (1.5 mm o.d., 0.86 mm i.d., Science Products GmbH), were pulled and polished to a final tip resistance of between 2 and 5 mOhm. Transfected cells are identified by expression of a co-transfected pRK5 plasmid encoding EGFP fluorescent signals or pIRES coexpression of fluorescent proteins. Currents were recorded from outside-out patches clamped at -60mV. Typical 20–80% peak rise times were between 0.3 and 0.5 ms, and recordings with a rise time above 0.6 ms were excluded from further analysis. To facilitate recordings of heteromeric responses, 20 μM IEM 1925 dihydrobromide (Tocris) was used to block currents contributed by GluA2-lacking receptors. Current–voltage (I/V) relationship of glutamate response was recorded and used to calculate the rectification index (RI) [$(I_{40mV} - I_{0mV})/(I_{-60mV} - I_{0mV})$]. Recordings from patches that displayed RI < 0.6 were excluded from further analysis for heteromeric receptor analysis. For Supplementary Fig. 11c, displayed RI was calculated as [$(I_{80 mV} - I_{0 mV})/(I_{-80 mV} - I_{0 mV})$] to maximise accuracy of measurement with the small magnitude of outward currents.

To determine entry into desensitization and steady-state currents, a 500 ms pulse of glutamate was applied, and multiple traces were recorded with an inter-sweep interval of 1 s. Rate of desensitization entry was measured by fitting a two-exponential function to the first 200 ms of desensitization decay, and the weighted time constant was calculated. Steady-state current was estimated as the percentage of current at 200 ms after peak. Recovery from desensitization was measured using a two-pulse protocol, consisting of a 500 ms pulse followed by a 10 ms pulse at increasing time interval (with increments of 10 ms). Recovery time course of TARP-associated AMPARs was first fitted with a Hodgkin-Huxley type equation and displayed an exponent value close to 1 suggesting a single rate-limiting transition step, as previously reported[3]. Consequently, all recovery rate was calculated using a monoexponential fit. Resensitization was calculated from a 5 s glutamate application, and represents the increase in current amplitude from steady-state (at 200 ms), as a percentage of peak current amplitude. Data visualization and statistical analysis were performed using GraphPad Prism.

Non-stationary fluctuation analysis (NSFA) was performed on the declining phase of macroscopic currents, evoked by application of 200 ms pulse of 10 mM glutamate on outside-out patches containing GluA1i (wild type or the I609A mutant) fused to TARPγ8 (as described[17]). The variance (σ2) of 10 to 100 current responses was calculated, grouped into ten amplitude bins, plotted against the mean current of the amplitude bin (Ī) and fitted with the parabolic function $σ^2 = iĪ - Ī^2/N - σ_o2$ to estimate the single-channel current amplitude (i), the total number of channels (N) and the background variance σ_o2. The current traces for NSFA were filtered at 10 kHz and sampled at 100 kHz.

**Protein expression, purification and cryo-EM grid preparation**. The GluA1/A2_γ8 complex was produced by transient transfection of A1_FLAG and A2_ γ8_eGFP encoding plasmids into HEK-Expi293F^TM cells. The purification of the sample was similar to the protocol described before[2], with minor alterations. Cells were lysed for 2 h in buffer containing 25 mM Tris, pH 8.0, 150 mM NaCl, 0.5 % (w/v) digitonin (Sigma), 5 μM NBQX (Tocris), 1 mM PMSF, 1× EDTA-free Protein Inhibitors (Roche). Insoluble material was removed by ultracentrifugation (131,000 × g, 45 min, 45–50 Ti rotor) and the lysate was incubated with anti-GFP beads for 3 h. Beads were washed with glyco-diosgenin (GDN) buffer containing

25 mM Tris, pH 8.0, 150 mM NaCl, 0.02% GDN (Anatrace). After protein elution by digestion with HRV 3 C protease, fractions were incubated with ANTI-FLAG M2 Affinity Gel (Sigma) for 1 h. Gel was washed twice with GDN buffer each time, and protein was eluted with 1 ml of 0.15 mg/ml 3× FLAG® peptide (Sigma) dissolved in GDN buffer. Protein was concentrated to 2.5 mg/ml and incubated with 100 µM cyclothiazide (CTZ, tocris) for 30 min prior to grid preparation. 2.6 µl of this CTZ-containing sample were quickly mixed with 0.4 µl of 0.75 M L-Glu, giving final concentrations of 100 mM L-Glu and 87 µM CTZ, and applied on glow-discharged holey carbon copper grids (Quantifoil Cu 0.6–1 300 mesh). Excess sample was blotted with filter paper for 3 s before plunge freezing in liquid ethane using a FEI Vitrobot Mark IV equilibrated at 4 °C and 100% humidity.

**Cryo-EM data collection and processing.** Data were collected on an FEI Titan Krios at the Diamond Light Source facility, equipped with an energy filter and a K3 camera. 9664 movies were collected in counting mode (50 frames, 4 s exposure, 51 e/Å$^2$ combined total dose) (Supplementary Fig. 1a). Magnification was ×81,000, resulting in a pixel size of 1.06 Å. Motion correction and CTF correction were performed with MotionCorr[4] and Gctf[5]. All subsequent steps were performed with RELION 3.1[6]. Particle picking was performed automatically, and particles were extracted and binned in a box of 80 px, resulting in a pixel size of 4.24 Å. After rounds of 2D classification, 3D classification was performed using the full-length A1A2γ8_NBQX map (EMD-4575) low-pass filtered at 30 Å as an initial model. 448,038 particles showing the AMPAR features were pooled together, extracted and binned in a box of 160 px (pixel size of 2.12 Å) and refined focusing on the TMD-LBD region in C2 symmetry. 3D classification without alignment allowed us to extract 90,524 open state particles at full pixel size and refined with a TMD-LBD mask in C2 symmetry, resulting in a map at 3.8 Å resolution after postprocessing. After CTF refinement, polishing and 3D refinement the resulting map had an overall resolution of 3.53 Å (FSC 0.143), with higher resolution in the TMD sector (extended Figs. 1, 2).

In parallel, the 448,038 particles were extracted at full pixel size and refined in C1 symmetry (TMD-LBD mask), and further classified without alignment. This classification allowed us to identify 155,176 particles showing desensitized LBDs. Focused refinement in C1 followed by postprocessing resulted in a desensitized map at 4.5 Å resolution (FSC 0.143). Further 3D classification without alignment allowed us to isolate a subset of 107,276 particles, which resulted in a map at 3.6 Å after refinement, polishing and postprocessing.

As two different maps had been obtained from two independent classifications, we compared the particles that had generated the maps and looked for duplicates. 8367 particles were present in both models. 3D refinement followed by 3D classification indicated that most of these duplicated particles had a desensitized conformation. These particles were included in the final desensitized subset and removed from the open-state particle subset. Using this strategy we isolated two independent particle datasets, the open dataset with 83,344 particles, and the desensitized dataset with 105,918 particles. 3D refinement in C2 symmetry was used to generate an open-state map of the TMD-LBD at 3.51 Å resolution (FSC 0.143). To further push the resolution, we created two masks covering either the LBD layer or the TMD layer and continued the 3D refinement. After postprocessing, the two obtained maps are at 3.39 Å (TMD) and 3.60 Å (LBD) resolution (FSC 0.143), respectively (Supplementary Figs. 1c, 2, 3). A composite map was generated using EMDA (https://www2.mrc-lmb.cam.ac.uk/groups/murshudov/content/emda/emda.html). A similar strategy was used for the desensitized state particles, obtaining three maps: the TMD-LBD map with an overall resolution of 3.57 Å resolution, the TMD map at 3.39 Å resolution and the LBD map at 4.78 Å resolution (FSC 0.143) (Supplementary Figs. 1c, 2, 3).

Model building and refinement were performed using Coot[7] and REFMAC5[8] within the CCP-EM software suite[9] and Phenix[10]. Initially, the TMD from the A1A2Y8_NBQX model (PDB: 6QKC) and a GluA2 LBD bound to cyclothiazide and L-Glu (PDB: 3TKD)[11] were rigid body fitted into the composite maps using Chimera[12]. Atomic B-factors were reset to 40 Å$^2$ prior to refinement. For manual refinement, both the EMDA maps and individual TMD and LBD maps were used (Supplementary Figs. 2, 3), while for restrained refinement in REFMAC5 or Phenix the TMD-LBD maps were used. Figures were prepared with Chimera or PyMOL[13]. The final models were validated using Coot and Molprobity[14]. Dimensions of the pore were calculated using the program HOLE within Coot[15].

**TARP-β1 loop classifications.** To analyse the TARP-β1 loop conformation, which were invisible in the high-resolution maps, we carried out 3D classifications using a mask covering two LBD protomers and the TARP γ8 extracellular region (Supplementary Fig. 6a). The 448,038 particles dataset (after 3D refinement applying C1 symmetry) was used for 3D classification without alignment. 3D classification resulted in two classes showing TARP-β1-GluA2-LBD ("B/D") interactions, two classes with TARP-β1-GluA1-LBD ("A/C") interactions, and one class with no clear interactions (Supplementary Fig. 6a). A similar strategy was used for the desensitized state dataset (105,918 particles) which resulted in classes with TARP-β1-GluA2-LBD ("B/D" interactions) (Supplementary Fig. 6b); the open state dataset (83,344 particles), which resulted in classes with TARP-β1-GluA1-LBD ("A/C") contacts or no contacts (Supplementary Fig. 6c); and the resting state (114,730 particles, reference), which showed both TARP-β1-GluA2-LBD and TARP-β1-GluA1-LBD contacts (Supplementary Fig. 6d). In the case of open and desensitized states it was evident that both TARP loops were showing different

conformations. Therefore a 3D classification was also carried out with a mask focused on the second loop, which showed no clear interactions for open and desensitized states (Supplementary Fig. 6b, c). To further improve the maps' quality in the loop region, we performed a sequential classification strategy starting from the 448,038 particles dataset. Particles corresponding to classes with TARP-β1-GluA2-LBD interactions were pooled together and refined in C1 symmetry, and the resulting map clearly showed desensitized LBDs (Supplementary Fig. 7a, left) as well as strong densities for the loops. Particles corresponding to classes with TARP-β1-GluA1-LBD interactions or no interactions resulted in maps reflecting open-states (Supplementary Fig. 7a, right). Additional 3D classifications without alignment using a mask around the loop with weaker signal were performed. Within the "B/D" pool there were some classes without clear interactions, while one class in the "A/C" pool had clear "B/D" contacts (Supplementary Fig. 7b). Refinement of the particles with mixed-loop features resulted in desensitized models with one β1 loop interacting with GluA2 D1 LBD while the other loop showed weaker signal. We merged together all the desensitized particles and all the open state particles and performed TMD-LBD refinements (Supplementary Fig. 7b). The refined desensitized model showed improved density in the loop region (Fig. 2b, Supplementary Fig. 7c). However, for the open state the β1 loop shows weaker densities (Supplementary Fig. 7d) and the contact between TARP-β1-GluA1-LBD appears more clearly after 3D classifications of the original refined model (Supplementary Fig. 6c), indicating that this contact is less stable.

**Molecular dynamics (MD) simulations.** For all resting state systems, MD was performed as described previously[53]. Two models were chosen from the models generated in that study, based on DOPE scores, and are called Model 1 and Model 2 here. Both models have an unrestrained, elongated β1-loop conformation, allowing the exploration of their conformational freedom. For more detailed interaction studies, further simulations would be necessary. However, these simulations are useful to interpret the β1-loop classifications results.

CHARMM-GUI v1.7[76] was used for the system setup. Since the role of glycosylation in TARP-AMPAR contacts is not clear, two systems were set up for each model: with and without glycosylation of the two TARP-γ8 glycosylation sites (N53 and N56). As glycosylation, two glycosaminoglycans were selected. As a result, four simulations were performed: Model 1 with glycosylation (Run 1), Model 1 without glycosylation (Run 2), Model 2 with glycosylation (Run 3), Model 2 without glycosylation (Run 4). A heterogenous asymmetric lipid composition was chosen as described before[53]. The solvent was TIP3P water[77] with 150 mM NaCl.

We also performed two simulations of the open state structure (runs 5 and 6). Missing loops were added using MODELLER[78], with restraints applied to the β1-loop to extend its conformation. 100 models were generated, and two models with the highest consensus DOPE score and SOAP-LOOP score were used for simulation. TARP residues N53 and N56 were glycosylated, and TARP E216 was protonated as it is surrounded by hydrophobic residues. Like the resting state, the open state systems were embedded in a heterogeneous, asymmetric lipid membrane. TIP3P water was used to solvate the system and 150 mM NaCl was added.

For all simulations, system equilibration was performed as recommended by CHARMM-GUI[76]. After minimization for 5000 steps, two equilibration steps in the NVT ensemble of 125 ps each, followed by four equilibration steps in the NPT ensemble of 125 ps or 500 ps, were performed, where harmonic restraints on the protein, and planar/dihedral constraints on the lipids were consecutively decreased. Following removal of all restraints, all simulations were run for 350 ns each, in the NPT ensemble. Resting state simulations were performed using GROMACS 2019.3[79] at a temperature of 310.15 K and a pressure of 1.0 bar. The Berendsen method[80] was used during equilibration for both temperature coupling (tau-t 1 ps) and pressure coupling (compressibility $4.5 \times 10^{-5}$ bar$^{-1}$ and tau-p 5 ps). Separate temperature coupling groups were used for protein, membrane and solvent, and pressure coupling was semi-isotropic. For production runs, the Nosé–Hoover temperature coupling method[81,82] was used with a tau-t of 1 ps and the semi-isotropic Parrinello–Rahman method[83] was used for pressure coupling with a compressibility of $4.5 \times 10^{-5}$ bar$^{-1}$ and a tau-p of 5 ps. The open state simulations were performed in NAMD 3.0[84] where the simulation temperature was controlled at 310 K by Langevin dynamics, with a damping coefficient of 1 ps$^{-1}$, and the pressure of the system was kept at 101.325 kPa (1.01325 bar or 1 atmosphere) using the Nosé–Hoover Langevin method with a piston period of 50 fs and piston oscillation decay time of 25 fs.

The CHARMM36m force-field[85] and a 2.0 fs time step for production runs was used for all systems. Analysis of the simulation data was carried out using CONAN[55] and VMD[86]. CONAN was used for analysing residue interactions, defined as having at least one atom pair within a distance of 5 Å. All analyses were performed with a sampling of 100 ps/frame.

**Reporting summary.** Further information on research design is available in the Nature Research Reporting Summary linked to this article.

## Data availability

The cryo-EM data generated in this study have been deposited in the EMDB under accession codes EMD-13969 (LBD-TMD), EMD-13970 (TMD), EMD-13971 (LBD) for open state GluA1/2_γ8; and EMD-13972 (LBD-TMD), EMD-13973 (TMD), EMD-

13974 (LBD) for desensitized GluA1/2_γ8. The structural models have been deposited in the PDB under accession codes 7QHB (LBD-TMD open state), and 7QHH (LBD-TMD desensitized state). Source data are provided with this paper.

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

## Acknowledgements

We thank Ondrej Cais for critical reading of the manuscript. We are grateful to LMB scientific computing and the EM facility for support, Paul Emsley for help with model building and Takanori Nakane for helpful comments with Relion 3.1. This work was supported by grants from the Medical Research Council (MC_U105174197) and BBSRC (BB/N002113/1) to I.H.G, and grants from the MCIN/AEI/ 10.13039/501100011033 and "ESF Investing in your future" to B.H (PID2019-106284GA-I00 and RYC2018-025720-I).

## Author contributions

I.H.G. supervised the study, and wrote the paper with input from B.H., J.-N.D., J.F.W. and J.M.K. D.Z. and B.K.K. performed protein purification and cryo-EM data collection, B.H. and B.K.K. performed data processing and model building. J.F.W., R.L. and H.H. designed and performed electrophysiological experiments, J.-N.D., J.M.K. and S.S. performed and analysed MD simulations.

## Competing interests

The authors declare no competing interests.

## Additional information



