## [Peer Review File · Nature Communications]

Mechanisms underlying TARP modulation of the GluA1/2- γ 8 AMPA receptorREVIEWER COMMENTS

Reviewer #1 (Remarks to the Author):

This manuscript reports systematic comparisons of the molecular models of heteromeric AMPAR-gamma8 complex in different functional states. The results provide important mechanistic insights into the isolated effect of g8 mediated modulation in contrast to a more complex (and physiological) setting when CNIH2 is co-present. The cryo-EM maps are at high quality, except for the beta1 loop. The MD simulation is used effectively to interpret the low-resolution features in the map. Key advances are (1) the new residue contacts that occur in the open channel state between g8 TM4 and M2 that controls modulation of rectification, (2) the identification of patch 1 and 2 of the LBD that contact g8 beta1 loop, (3) the structural coupling of the Q/R site side chain and M3 during the gating cycle, (4) the differential positioning of the ion-like densities in the pore between conformations, (5) the rearrangement of side chain contacts at the apex of the M3 and the surrounding pre-M1 collar upon gate opening in comparison to when CNIH2 is present, and (6) the hypothesis that charges in the g8 beta4 loop funnels ions to the pore. Important functional models deduced from the structures are tested by recording from mutant channel complexes with key residues interrogated. These new mechanistic insights are critical in understanding the mechanism for the AMPAR gating modulation by TARPs. Overall, this work is an important contribution to the field of molecular neuroscience and ion channel structural biology. The following are some suggestions.

Major points:

Were there any reconstructions with one CTZ, which may represent partially desensitized structures? What I am wondering is if there is a structure with only one LBD dimer desensitized.

Providing the resolutions, global and local, of the maps in Extended Fig 7 would be informative to evaluate the map quality. Please also provide the angular distribution of the particles contributing to these reconstructions. In the starting map of C1 symmetry, there are two sides (front and back) to consider in choosing the mask of the LBD and beta1 loop. What does the map look like if you refine (or classify) the rear side with an appropriate mask? It is curious how much asymmetry exists in the beta1 loop interaction with the LBDs?

The angular distribution of the particles is skewed to the top views. It appears as if the majority of the particles with the side views (shown as class averages in Extended Fig1b) were removed during the 3D alignment and refinement (Extended Fig2). The skewed distribution does not appear to have introduced concerning artifacts, but it is puzzling where all of the side views in the 2D class averages disappeared. Some comments on this point in the methods would be helpful.

The authors demonstrate the residue contacts between gamma-8 TM4 and M2 helix, and further show functional significance of the contact in modulating rectification. The residue GluA1 D586 on the cytoplasmic side of the filter loop was shown to have functional significance in the rectification (Soto et al JN 2009). Is this residue altered in anyway in the presence of gamma-8 in the author's structure?

The flexible beta1 loop of gamma-8 is somewhat visible, which appears to be at relatively low resolution. The authors did an excellent job collecting a large dataset to partially resolve the loop. The molecular interpretation of the peripheral densities of the loop, however, relies heavily on the MD simulations, and leaves questions about whether the two orthogonal approaches (cryo-EM and MD) are producing coherent models. The Patch 1 and 2 on the LBD are interrogated by mutations and the result makes a convincing case. The hypothesis would be further strengthened by recording from the complexes with mutations in the beta1 loop that are predicted by MD to make contact with the two LBD patches.

In the methods, the boundary of the beta1-loop chimera made between gamma-2 and gamma-8 should be provided with exact the amino acid sequences of the junction. The loops are divergent among the TARPs and small changes may have functional consequences. It is interesting that the authors find that only the recovery from desensitization was affected by exchanging the loop. Examining how the chimeric loop behaves in MD simulation in comparison to the wild-type gamma-8 might help identify residues in the beta1 loops that contribute to modulating recovery.

Minor issues:

Scales for the vectors that indicate the motion should be provided in Extended Fig 5. The definition for center of rotation for the LBD should be provided in the methods or in the legend on Extended Fig 5.

Extended Fig 12d: The labels are placed in a confusing way. The locations of labels for GluA1 and TARPg-8 could be exchanged. The lipid label might be clearer, if placed on the bottom.

Reviewer #2 (Remarks to the Author):

The manuscript by Herguedas et al. presents a combined study using cryo-electron microscopy, MD simulations and patch clamp electrophysiology to investigate the role of TARPs in AMPA gating and conductance. The presented cryo-EM structure is of high quality and the overall finding is robust and novel. I have nevertheless some minor issues mainly related to the MD simulation part.

1. The authors discussed about the MD simulations of GluA1/2-gamma8 complexes. It would be very useful for the readers, if the authors could provide more details about MD simulations in the main text, e.g. How long are the simulations altogether? What were the starting structures of these simulations? Do the simulations of resting state and open state show similar interaction profile between LBD and gamma-8?

2. The authors described a coupling between the gate and filter entrance via Ile613? Did the authors perform additional mutation studies to confirm the role of Ile613?

Reviewer #3 (Remarks to the Author):

The manuscript by Herguedas describes the resolution of GluA1/2 heteromeric AMPA receptor complexes containing the gamma-8 TARP in open and desensitized states. The new information in this report complements two recent studies published in Nature, one from the same research group, which described GluA1/2 with gamma-8 and CNIH2 (Zhang 2021) and native hippocampal AMPA receptors with TARPs, CNIH2, and SynDIG4 (Yu 2021). It also represents an extension of another study with the receptor complex from the same laboratory in 2019. The data is of high quality as is typical from the Greger laboratory, and the structures are analyzed and discussed rigorously.

The key assessment for the current report whether it contributes new, substantive insights to understanding of TARP modulation of AMPA receptor gating. The Introduction does an inadequate job of setting up the context for this study, unfortunately. The Results and Discussion are much more informative in that regard. The data refines some of the molecular contacts described in previous structures and extend the resolution of contacts between subunits and auxiliary proteins. The take-away messages from this and the earlier 2021 study appear very similar. For example, the last paragraph of the Herguedas Introduction describes, "gating-state-specific contacts of the extracellular γ 8 loops with the LBDs and gating linkers", which also was reported in the Zhang 2021 Nature study ("TARPs engage the LBDs and the M1 and M3 gating linkers through their extracellular portion", last paragraph). The current report takes a similar approach but leaves out the CNIH2 protein, which the Yu paper suggests is a common constituent of native AMPA receptors complexes. One is left wanting an explanation from the authors why these results yield a new understanding of TARP modulation of gating because they are inadequately contextualized with existing literature and structures.

Minor criticism

The authors state that the TARPs, “modulate gating through currently unresolved mechanisms”, and cite studies which arguably report the very thing they claim has not been done. As well, one might note that the Ben-Yaacov 2017 paper is titled, “Molecular Mechanism of AMPA Receptor Modulation by TARP/Stargazin”, which could reasonably be interpreted as providing mechanistic insight into modulation of gating by auxiliary subunits. It is difficult to understand why the authors felt obliged to make this sort of sweeping statement given the plethora of contradictory reports. Perhaps they had in mind a narrower meaning; if so, more precise wording is recommended.

Response to reviewers Nat Comm. -21-30995-T (Herguedas et al.)

Reviewer #1:

This manuscript reports systematic comparisons of the molecular models of heteromeric AMPAR-gamma8 complex in different functional states. The results provide important mechanistic insights into the isolated effect of g8 mediated modulation in contrast to a more complex (and physiological) setting when CNIH2 is co-present. The cryo-EM maps are at high quality, except for the beta1 loop. The MD simulation is used effectively to interpret the low-resolution features in the map. Key advances are (1) the new residue contacts that occur in the open channel state between g8 TM4 and M2 that controls modulation of rectification, (2) the identification of patch 1 and 2 of the LBD that contact g8 beta1 loop, (3) the structural coupling of the Q/R site side chain and M3 during the gating cycle, (4) the differential positioning of the ion-like densities in the pore between conformations, (5) the rearrangement of side chain contacts at the apex of the M3 and the surrounding pre-M1 collar upon gate opening in comparison to when CNIH2 is present, and (6) the hypothesis that charges in the g8 beta4 loop funnels ions to the pore. Important functional models deduced from the structures are tested by recording from mutant channel complexes with key residues interrogated. These new mechanistic insights are critical in understanding the mechanism for the AMPAR gating modulation by TARPs. Overall, this work is an important contribution to the field of molecular neuroscience and ion channel structural biology. The following are some suggestions.

We thank the reviewer for the positive feedback on our work.

Major points:

- Were there any reconstructions with one CTZ, which may represent partially desensitized structures? What I am wondering is if there is a structure with only one LBD dimer desensitized.

We did not obtain any reconstruction of sufficient quality that could represent a partially/semi desensitised model. The dataset containing 448K particles, included ~ 250K particles that belonged to neither the open or desensitized state classes. 3D refinement of this subset of particles resulted in a reconstruction with an overall resolution of 8.92 Å (please see Fig. 1R below), which could correspond to a partially desensitised model, with the two LBD dimers showing different conformations. Further classification or focused refinement did unfortunately not improve its quality. Due to its low resolution, we cannot conclude whether this reconstruction is a mixture of open, desensitised and poor-quality particles, or a partially desensitised receptor.

Fig. 1R:

-Providing the resolutions, global and local, of the maps in Extended Fig 7 would be informative to evaluate the map quality. Please also provide the angular distribution of the particles contributing to these reconstructions. In the starting map of C1 symmetry, there are two sides (front and back) to consider in choosing the mask of the LBD and beta1 loop. What is does the map look like if you refine (or classify) the rear side with an appropriate mask? It is curious how much asymmetry exists in the beta1 loop interaction with the LBDs?

In the previous version of the manuscript we did not show the resolution (global or local) or particle distribution of the classes as they were the result of a classification without alignment using a mask covering the loop. To obtain these, we have now refined models corresponding to either "BD"-interactions or "AC+ no" interactions, as requested (revised Extended Data Fig. 6 and 7). Interestingly, and in support of our conclusions, the BD particles selectively generated desensitized models while the AC+no int. particles generated open state models. We have now expanded our discussion on this point and present two new figures, Extended Data Fig. 6 and 7, to document the processing strategy of each of these refined models (including local resolution and particle distribution). Moreover, focused classifications on each of these refined models using the 'loop mask' revealed that some particles initially classified as "AC+ no int" or "BD" contained some particles with the alternative loop interaction. Therefore, we performed a hierarchical classification where 3D-refinement was followed by 3D classification without alignment and then pooled classes with similar loop features. Using this strategy we obtained two new models, a "BD" model, showing a clear desensitized conformation and a strong BD-loop interaction, and an "AC" model, which shows an open-state conformation with an AC-loop interaction. The BD interaction is stronger than the AC interaction, indicating that the BD interaction is more frequent in the desensitized state than the AC interaction in the open-state.

Finally, regarding asymmetry of the loop, we have observed asymmetry in all our 3D-refinements, with only one of the two loops showing a clear interaction with the LBDs. This is particularly relevant for BD LBD interactions, where the signal of the loop is relatively strong for one of the two loops, while the other is almost absent). We comment on these observations in the revised text (page 9).

- The angular distribution of the particles is skewed to the top views. It appears as if the majority of the particles with the side views (shown as class averages in Extended Fig1b) were removed during the 3D alignment and refinement (Extended Fig2). The skewed distribution does not appear to have introduced concerning artifacts, but it is puzzling where

all of the side views in the 2D class averages disappeared. Some comments on this point in the methods would be helpful.

Thank you for pointing this out. To clarify, the 2D classes shown in Extended Fig. 1b were generated with particles from open and desensitised states, as we did not perform a 2D classification with un-binned particles. Therefore, the side views shown in this figure are actually present in the refined models. However, the previous version of Extended Figure 1b was showing only the first 48 classes of the 2D classification, and not all the classes, and therefore it seemed that side views were more abundant. In the revised Extended Figure 1b, we present all classes obtained after 2D classification of open+desensitised state particles. We have also analysed the particle distribution in the early stages of processing, and an excess of top views is observed at all stages of processing. Below, we show the 2D classifications of (1) Open and desensitised state particles (Fig 2R; now in Extended Figure 1B), (2) 448,000 dataset Fig 3R, and (3) Particle distribution after 3D refinement of 448K particles, where top views were also abundant (Fig 4R).

Fig. 2R: 2D classification, desensitised + open state particles

Fig. 3R: 2D classification, complete dataset after cleaning (448 K particles)

Fig. 4R: Particle distribution of the map obtained after 3D refinement of the full dataset (448K particles)

The authors demonstrate the residue contacts between gamma-8 TM4 and M2 helix, and further show functional significance of the contact in modulating rectification. The residue GluA1 D586 on the cytoplasmic side of the filter loop was shown to have functional significance in the rectification (Soto et al JN 2009). Is this residue altered in anyway in the presence of gamma-8 in the author's structure?

As the reviewer points out, this Asp residue at the cytoplasmic pore entrance is essential for polyamine binding. The Asp side chain conformations are comparable between our current open and desensitized structures arguing against state-dependent changes. When compared against our recent (GluA1/GluA2 TARP γ 8) resting state structure (PDB 7OCD), we see

reorientation of the Asp side chains in the GluA2 subunits, but do not observe any other local alterations correlating with this change. Overall, there are currently no clear-cut state dependent changes at this site.

The flexible beta1 loop of $\gamma 8$ is somewhat visible, which appears to be at relatively low resolution. The authors did an excellent job collecting a large dataset to partially resolve the loop. The molecular interpretation of the peripheral densities of the loop, however, relies heavily on the MD simulations, and leaves questions about whether the two orthogonal approaches (cryo-EM and MD) are producing coherent models. The Patch 1 and 2 on the LBD are interrogated by mutations and the result makes a convincing case. The hypothesis would be further strengthened by recording from the complexes with mutations in the beta1 loop that are predicted by MD to make contact with the two LBD patches.

As outlined in response to major point 2 (page 2) we have now conducted additional cryo-EM data processing, which further substantiated our initial observations (revised paper, Ext. Data Fig. 6 and 7). Namely, the beta1 loop selectively engages GluA2 (patch 1) under desensitizing conditions, and GluA1 (patch 2) in both resting and open states. We also obtained better maps as shown in Ext. Data Fig. 7b and c of the revision. As outlined below, to further address this point we also conducted new MD simulations of the open state, where beta-1 loop interactions with patch 2 prevail (revised paper, Ext. Data Fig. 7b, c). We feel that this additional new evidence further strengthens the major point of this results section and obviates the need for further recordings.

In the methods, the boundary of the beta1-loop chimera made between gamma-2 and gamma-8 should be provided with exact the amino acid sequences of the junction. The loops are divergent among the TARPs and small changes may have functional consequences.

Thank you for pointing this out. This information is now provided in the Methods section on p. 23: 'The TARP $\gamma 8$ $\beta 1$ -loop chimera was constructed by replacing $\gamma 8$ residues Leu50-Leu82, with the equivalent region of $\gamma 2$ Val39-Met58.'

It is interesting that the authors find that only the recovery from desensitization was affected by exchanging the loop. Examining how the chimeric loop behaves in MD simulation in comparison to the wild-type $\gamma 8$ might help identify residues in the beta1 loops that contribute to modulating recovery.

We had also considered this interesting point and had simulated the gamma8/gamma2 loop chimera, the trajectories of which are shown below (Fig. 5R). The chimeric loop can interact with the same LBD patches (Fig. 2 of the revised paper), but the dynamics and interaction range are different compared to the gamma 8 loop. However, to obtain the required sampling for an in-depth analysis, substantially more interaction events would be required, before interacting residues can be determined with some accuracy. This would require very lengthy simulations, which we feel are not justified.

Fig. 5R: 350 ns simulations of either TARP $\gamma 8$ or the loop chimera (Ex1 of $\gamma 2$). Shown are overlays of the receptors every 1 ns of the entire trajectory. TARP $\gamma 8$ (green), GluA1 LBD (blue), GluA2 LBD (red).

Minor issues:

- Scales for the vectors that indicate the motion should be provided in Extended Fig 5. The definition for center of rotation for the LBD should be provided in the methods or in the legend on Extended Fig 5.

Thank you, we have now added this information in the legend of Ext. Data Fig. 4a, as follows:

The degree of rotation undergone by the C α atoms around a centre of rotation of an LBD dimer is shown for each transition by colouring the starting structure from white (least mobile) to red (most mobile): closed to open (left), open to desensitised (middle), desensitised to closed (right). The centre of rotation was inspired by the vectors shown in panel b, and were chosen as the centre of mass from either an LBD dimer (closed to open and desensitised to closed transitions), or an LBD monomer (open to desensitised transition). The LBD was defined by the following residues GluA1:392-502, 630-769; GluA2: 396-506, 634-773.

Extended Fig 12d: The labels are placed in a confusing way. The locations of labels for GluA1 and TARP γ -8 could be exchanged. The lipid label might be clearer, if placed on the bottom.

We agree with the reviewer and have altered the labelling as requested.

Reviewer #2 (Remarks to the Author):

The manuscript by Herguedas et al. presents a combined study using cryo-electron microscopy, MD simulations and patch clamp electrophysiology to investigate the role of TARPs in AMPA gating and conductance. The presented cryo-EM structure is of high quality and the overall finding is robust and novel. I have nevertheless some minor issues mainly related to the MD simulation part.

We thank the reviewer for their positive comments on our work.

1. The authors discussed about the MD simulations of GluA1/2-gamma8 complexes. It would be very useful for the readers, if the authors could provide more details about MD simulations in the main text, e.g. How long are the simulations altogether? What were the starting structures of these simulations? Do the simulations of resting state and open state show similar interaction profile between LBD and gamma-8?

We thank the reviewer for pointing this out. Each simulation was 350 ns long, and the combined simulation time for the resting state is 1400 ns, while that for the open state is 700 ns. We have now included the necessary details for MD simulations in the main text (page 10), including a new interaction profile for the open state in Ext. Data Fig. 8b, c.

2. The authors described a coupling between the gate and filter entrance via Ile613? Did the authors perform additional mutation studies to confirm the role of Ile613?

That is a very interesting point. We have now conducted functional experiments, to directly assay coupling between the gate and the selectivity filter. In particular, we show the GluA1 I609A mutation increases conductance of the channel, as predicted from our structures. This can be explained by a widening of the Q582 side chains in the absence of the bulky isoleucine (I609) projecting from the M3 helix. These data are outlined on pages 13/14 and are presented in Fig. 4c of the revised paper. We conducted similar experiments with the analogous GluA2 mutant (I613A) but the responses were too small for accurate measurement by non-stationary noise analysis, as we describe in the revised text (pages 13/14).

Reviewer #3 (Remarks to the Author):

The manuscript by Herguedas describes the resolution of GluA1/2 heteromeric AMPA receptor complexes containing the gamma-8 TARP in open and desensitized states. The new information in this report complements two recent studies published in Nature, one from the same research group, which described GluA1/2 with gamma-8 and CNIH2 (Zhang 2021) and native hippocampal AMPA receptors with TARPs, CNIH2, and SynDIG4 (Yu 2021). It is also represents an extension of another study with the receptor complex from the same laboratory in 2019. The data is of high quality as is typical from the Greger laboratory, and the structures are analyzed and discussed rigorously.

We thank the reviewer for their positive comments.

The key assessment for the current report whether it contributes new, substantive insights to understanding of TARP modulation of AMPA receptor gating. The Introduction does an inadequate job of setting up the context for this study, unfortunately. The Results and Discussion are much more informative in that regard.

Following the advice of the reviewer, we have revised the introduction and hope that this has improved clarity.

The data refines some of the molecular contacts described in previous structures and extend the resolution of contacts between subunits and auxiliary proteins. The take-away messages from this and the earlier 2021 study appear very similar. For example, the last paragraph of the Herguedas Introduction describes, “gating-state-specific contacts of the extracellular $\gamma 8$ loops with the LBDs and gating linkers”, which also was reported in the Zhang 2021 Nature study (“TARPs engage the LBDs and the M1 and M3 gating linkers through their extracellular portion”, last paragraph). The current report takes a similar approach but leaves out the CNIH2 protein, which the Yu paper suggests is a common constituent of native AMPA receptors complexes. One is left wanting an explanation from the authors why these results yield a new understanding of TARP modulation of gating because they are inadequately contextualized with existing literature and structures.

The reviewer queries the novelty of the current study and asks how it extends beyond our recent work (Zhang et al. Nature 2021) and that of others (Yu et al., Nature 2021). To address this, we have revised the text throughout, to clarify how the paper contributes ‘new understanding to TARP modulation of gating’. Our current study features a number of key mechanistic findings that are not covered in these two Nature papers, or elsewhere in the literature. Some of these advances were also summarized by reviewer 1, as stated in their review:

- 1. The TARP- $\gamma 8$ TM4/GluA1 M1 interaction (V220-I569) regulating rectification (Fig. 5)*
- 2. Identification of TARP- $\gamma 8$ beta-1 loop interaction patches on the LBD (Fig. 2)*
- 3. Coupling between the selectivity filter and the M3 gate through the Q/R site (Fig. 4a-c)*
- 4. State-dependent ion densities in the conduction path (Fig. 4d,e and Extended Data Fig. 10).*
- 5. Comparison of gate contacts in open states between $\gamma 8$ -only versus $\gamma 8$ +CNIH2 structure (Fig. 3 and Ext. Data Fig. 11)*
- 6. A potential role of the $\gamma 8$ beta-4 (‘acidic’) loop in funneling cations to the pore entry (Ext. Data Fig. 9c,d).*

As pointed out, none of these insights appear in the two 2021 Nature papers.

Importantly, in point 2 (of the above list) we reveal state-dependent interactions between the beta-1 loop and the LBD. These are supported further by newly conducted MD simulations of the open state (page 10 and Ext. Data Fig. 8b,c of the revision) and by additional processing of the cryo-EM data (Ext. Data Fig. 6 and 7). Together, we provide 1st insights into how a Type-1 TARP engages the receptor during its gating cycle.

With regard to point 3, we have now extended the original structural observations with electrophysiological recordings, demonstrating a functional connection between the isoleucine side chains in the M3 gating helices with the Q/R site at the selectivity filter apex. Coupling between the two major ion constriction elements (gate and selectivity filter) is entirely novel (Fig. 4).

In addition to these points, we provide a TMD/ion channel structure of a desensitized state at ~ 3.5 angstrom, permitting direct comparisons to open and resting state receptors of the same AMPAR complex (GluA1/2 associated with two $\gamma 8$ subunits), enabling in-depth functional characterisations in future studies.

Minor criticism

The authors state that the TARPs, “modulate gating through currently unresolved

mechanisms”, and cite studies which arguably report the very thing they claim has not been done. As well, one might note that the Ben-Yaacov 2017 paper is titled, “Molecular Mechanism of AMPA Receptor Modulation by TARP/Stargazin”, which could reasonably be interpreted as providing mechanistic insight into modulation of gating by auxiliary subunits. It is difficult to understand why the authors felt obliged to make this sort of sweeping statement given the plethora of contradictory reports. Perhaps they had in mind a narrower meaning; if so, more precise wording is recommended.

While we agree with the reviewer that some mechanistic insights into TARP action exist, the field is still far from understanding the mechanisms underlying allosteric coupling between TARPs (and any other auxiliary subunit) with the AMPAR. These mechanisms will be essential for understanding how TARPs tune kinetics, hence charge transfer through the AMPA channel, and ultimately synaptic computations. TARPs have now been studied for ~ 20 years, and therefore an extensive list of biophysical changes that TARPs confer onto AMPA receptors, including gating kinetics, ion conductance, rectification by polyamines (intra- and extra-cellular) and effects on AMPAR pharmacology, has been compiled. Some of this is covered in our 2017 Neuron review (Greger et al., 2017). Most of these insights come from mutagenesis combined with functional data - but how these observations actually translate into structural changes, and how these are manifested in the complex AMPAR gating cycle is far from clear. Therefore, our mechanistic understanding of TARP action is currently rudimentary. Obtaining this missing information will require more complete structures that also fully resolve missing elements such as the gating linkers (including side chains), TARP (extracellular) and receptor (intracellular) loops and their C-termini. Resolving AMPAR combinations at different states of the gating cycle, and at better resolution (in the low 2 angstrom range) will be required. The field has not reached this stage but some of these questions are starting to be addressed in our current study.

I'd also like to clarify that the above-mentioned Ben-Yaacov study revealed that TARPs associate with the AMPAR M1 and M4 transmembrane helices, which confirmed early AMPAR-TARP structures from the Gouaux and Sobolevsky labs. But, as I pointed out above, how this association transmits functional changes is far from clear. and was not addressed in these studies.

Lastly, in the abstract we have changed “modulate gating through currently unresolved mechanisms” to “modulate gating through currently incompletely resolved mechanisms” to address the reviewer’s point. We have also kept their comments in mind when revising the remainder of the paper.

REVIEWERS' COMMENTS

Reviewer #1 (Remarks to the Author):

The authors addressed the reviewers' critiques in the revised manuscript. I have two comments, which is up to the authors to decide whether to incorporate into their final version.

1. I recommend the authors to confirm if the residues in the Q/R site are the sole interacting partners of the side chain of I609. I609 might also have van der Waals contact with additional residues in M2. If this is the case the interpretation of the recording phenotype of I609A mutation would require caution.

2. Line 313: This should be corrected to “hydrated Na⁺ ions”, considering that the Pauling radius of Na⁺ is 0.95Å.

Overall, this a high quality work that advances our understanding of the mechanism of AMPAR modulation by TARPs.

Reviewer #2 (Remarks to the Author):

The authors have addressed all my questions and comments properly.

Reviewer #3 (Remarks to the Author):

Authors effectively clarified the import of new findings in their study. No further concerns.

Response to reviewers NCOMMS-21-30995A; Herguedas et al.

We thank the reviewers for their feedback. We have addressed the 2 remaining comments raised by reviewer 1 below.

Reviewer #1 (Remarks to the Author):

The authors addressed the reviewers' critiques in the revised manuscript. I have two comments, which is up to the authors to decide whether to incorporate into their final version.

1. I recommend the authors to confirm if the residues in the Q/R site are the sole interacting partners of the side chain of I609. I609 might also have van der Waals contact with additional residues in M2. If this is the case the interpretation of the recording phenotype of I609A mutation would require caution.

We agree with the reviewer that a Phe side chain in M2 is in van der Waals distance of the M3 isoleucines, I609 (GluA1) and I613 (GluA2), but are slightly farther than the Q/R residues. We now include the following on p. 14; line 292: 'Shortening the M3 isoleucine side chain is expected to alleviate interaction with the M2 pore loop apex, primarily through the Q/R site residues, and to a lesser extent with GluA1/GluA2 residues F580/F584.'

2. Line 313: This should be corrected to "hydrated Na⁺ ions", considering that the Pauling radius of Na⁺ is 0.95Å.

We have included this as advised on p. 14; line 314.

Overall, this a high quality work that advances our understanding of the mechanism of AMPAR modulation by TARPs.

We thank the reviewer for the 2 remaining comments, which we have included in the revised text.

No other comments were raised:

Reviewer #2 (Remarks to the Author):

The authors have addressed all my questions and comments properly.

Reviewer #3 (Remarks to the Author):

Authors effectively clarified the import of new findings in their study. No further concerns.